# Association between Eating Patterns and Quality of Life in Patients with Familial Hypercholesterolemia

**DOI:** 10.3390/nu15163666

**Published:** 2023-08-21

**Authors:** Alexandra Maștaleru, Irina Mihaela Abdulan, Andra Oancea, Alexandru Dan Costache, Raul-Alexandru Jigoranu, Mădălina Ioana Zota, Mihai Roca, Ileana-Katerina Ioniuc, Cristina Rusu, Laura Mihaela Trandafir, Elena Țarcă, Maria Magdalena Leon, Carmen Marinela Cumpăt, Florin Mitu

**Affiliations:** 1Department of Medical Specialties I, “Grigore T. Popa” University of Medicine and Pharmacy, 700115 Iasi, Romania; alexandra.mastaleru@gmail.com (A.M.); adcostache@yahoo.com (A.D.C.); jigoranu_raul-alexandru@d.umfiasi.ro (R.-A.J.); madalina.chiorescu@gmail.com (M.I.Z.); roca2m@yahoo.com (M.R.); leon_mariamagdalena@yahoo.com (M.M.L.); marinela.cumpat@umfiasi.ro (C.M.C.); mitu.florin@yahoo.com (F.M.); 2Clinical Rehabilitation Hospital, 700661 Iasi, Romania; 3Department of Mother and Child, “Grigore T. Popa” University of Medicine and Pharmacy, 700115 Iasi, Romania; ileanaioniuc@yahoo.com (I.-K.I.); abcrusu@gmail.com (C.R.); trandafirlaura@yahoo.com (L.M.T.); 4Department of Surgery II—Pediatric Surgery, “Grigore T. Popa” University of Medicine and Pharmacy, 700115 Iasi, Romania; elatarca@gmail.com

**Keywords:** familial hypercholesterolemia, overeating, quality of life, SF-36 questionnaire, mental status

## Abstract

(1) Background: Familial hypercholesterolemia (FH) is a genetic disease that has autosomal dominant inheritance, being characterized by increased levels of low-density lipoproteins (LDLs) due to a decreased clearance of the circulant LDLs. Alimentation is a key factor in patients with FH. Implementing a restrictive diet may have a significant impact on their quality of life, besides the social and environmental factors. (2) Methods: We realized a prospective study that was conducted in the Cardiovascular Rehabilitation Clinic from the Clinical Rehabilitation Hospital and that included 70 patients with FH and 20 controls (adults with no comorbidities). We evaluated their lipid profile, their quality of life through the Short Form—36 Questionnaire, and their eating habits. (3) Results: Lower scores in the quality-of-life questionnaire were obtained in the FH group both in the case of the physical (73.06 vs. 87.62) and the mental component (75.95 vs. 83.10). Women had better physical function (85 vs. 75) and physical role than men (100 vs. 75). The group aged over 65 has the score lowest for all 10 components. Overeating was driven by boredom and was more frequent on weekends in the FH group. None of the patients in the control group felt loneliness or depression associated with overeating. (4) Conclusions: Overeating in patients with FH is associated with a lower quality of life. The complexity of these patients needs a multidisciplinary approach. Thus, the quality-of-life questionnaire should be implemented in their periodic follow-ups in order to increase their general status, paying special attention to geriatric patients.

## 1. Introduction

Familial hypercholesterolemia (FH) is a genetic disease with autosomal dominant inheritance, characterized by increased levels of low-density lipoproteins (LDLs) due to a decreased clearance of the circulant LDLs. In time, they will deposit on the arterial walls, causing a significant increase in the risk of developing premature cardiovascular disease [1].

FH can have two genetic forms: homozygous and heterozygous. Meta-analyses using clinical definitions of FH with the inclusion of increasing numbers of patients have established a global prevalence for the heterozygous form of approximately 1:300, with considerable variability between regions, despite the fact that some populations are not included currently in studies. Regarding Europe, a meta-analysis that included more than 80% of cases from the general European population estimated the prevalence to be 1:311, ranging between 1:200 and 1:575 [2]. The homozygous form, however, is much rarer, with a prevalence of 1:170,000–1:300,000 [3].

The most frequent causes of FH are mutations in the LDLR gene (approximately 90%), followed by a mutation in the APOB (about 10%), while less than 5% can have a defect of the PCSK9 gene. Extremely rarely, less than 1% can have a defect in the LDLRAP1 gene [4].

Several factors contribute to blood cholesterol levels, such as diet, ethnicity, and family history. Nevertheless, an unhealthy diet is a risk factor for hypercholesterolemia that can be modified. Adopting a balanced diet low in saturated fats is a crucial measure individuals can take to decrease their chances of developing high blood cholesterol [5,6]. This aspect is also valid for patients with FH, where not only the genetic factor is involved.

But when eating habits turn into potential causes in the development or exacerbation of metabolic pathologies, an increased interest is placed on overeating, its causes, and its effects.

The lack of nutritional education programs, information on the harmful effects of a correct diet, as well as the interventions of psychologists on the impact that a genetic disease has on patients are direct causes of the inconsistent management of patients with FH.

The purpose of this research is to find out the effects of nutrition on the quality of life of this category of patients.

## 2. Materials and Methods

### 2.1. Type of Study

We realized a prospective study that was conducted in the Cardiovascular Rehabilitation Clinic from the Clinical Rehabilitation Hospital between 1 December 2020 and 31 March 2022. The study included 70 patients with familial hypercholesterolemia, a diagnosis that was based on a score of more than 8 points in the Dutch Lipid Clinic Network (DLCN). Because many of the patients had already had, in their daily medical treatment, lipid-lowering therapy, we corrected their LDL cholesterol value with a factor that was described by Haralambos et al. [7]. The value was corrected only for the patients that had administered a lipid-lowering treatment in the past six months, and for the others, the included LDL cholesterol value was the one obtained by our laboratory.

### 2.2. Patients’ Selection

During the aforementioned period, we identified, from a total of 2005 patients admitted to our clinic, 70 with FH who met the inclusion criteria to participate in the present study and 20 patients in the control group, patients with normal lipid profile values.

The inclusion criteria were a DLCN score of more than 8 points, an age over 18 years old, and a signing of the informed consent. The exclusion criteria were a DLCN score of less than 8 points and the diagnosis of at least one of the following conditions: hypothyroidism, chronic kidney disease with a low creatinine clearance, nephrotic syndrome, diabetes mellitus with uncontrolled glycemic values, severe liver disease, or a hypercaloric diet (total fat intake more than 40% of the daily caloric intake) [8], in order to diminish the possibility of secondary hypercholesterolemia (Figure 1).

### 2.3. Data Collection

A single investigator performed both history and physical examinations and collected general information (sex and age).

Eating habits were evaluated by applying the food frequency questionnaire. It includes information related to the number of daily meals, the number of snacks, favorite foods, the types of fats consumed or used in cooking, and the factors determining overeating and its effects.

Quality of life was studied using the Short Form Questionnaire 36 (SF-36). It is probably the most widely used tool for measuring the quality of life. It includes 36 items and was designed to provide a comprehensive assessment of the physical, mental, and social components of health status. More specifically, it uses 8 scales: physical function, social function, physical role limitation due to physical or emotional causes, mental health, energy level, somatic pain, and general health. The interpretation is complex: a scoring of the 36 answers, then 8 scales unify the items, finally obtaining a score consisting of the score of the two concepts—physical and mental. The mentioned scales are scored from 0 to 100. High scores indicate a better quality of life.

### 2.4. Ethics Committee

In order to be enrolled in the study, all included patients signed the informed consent. The study was approved by the Ethics Committee of both the “Grigore T. Popa” University of Medicine and Pharmacy Iași (15 June 2020) and the Iași Clinical Rehabilitation Hospital (25 November 2020).

### 2.5. Statistical Analysis

Statistical analysis was performed with SPSS 20.0 (Statistical Package for the Social Sciences, Chicago, IL, USA). Categorical variables were expressed as percentages and were compared by the Chi-Square test. The Shapiro–Wilk test was applied to assess the normality of distribution for continuous variables. The variables with a normal distribution were presented as mean values with standard deviations and were compared by the Student’s *t*-test. Non-normally distributed continuous variables were presented as medians with interquartile ranges and were compared with the Mann–Whitney U test. The threshold of statistical significance was a value of *p* ≤ 0.05 for all analyses.

## 3. Results

Table 1 shows the general characteristics of the study groups. There were no statistically significant differences between the two groups in terms of gender distribution and living environment, but the mean age of the FH patients was significantly higher than that of the control group.

The quality-of-life assessment reveals numerous statistically significant differences. We observe higher scores of patients in the control group, both in the physical and mental components (Table 2).

According to each dimension of the SF-36 questionnaire, the control group scores statistically significantly higher on almost all items (Table 3).

Even in the case of statistically insignificant results, the group of patients with FH presents lower scores, thus implying a lower quality of life.

### 3.1. General Characteristics

We found no statistically significant differences in terms of gender and quality of life in patients with FH, but we note that women had better physical function and physical role than men, and somatic pain, overall health, vitality, and social function had higher scores among males. Moreover, we note that physical function was better among rural people, but general and mental health, physical role, and social function had better scores in urban patients. Evaluating somatic pain, it was more intensely perceived by people from rural areas compared to those from urban areas (*p* = 0.002) (Table 4).

Significant gender differences were observed among patients included in the control group for the physical component and social function, with males having higher scores than females. Without significant differences, we mention that men presented a higher vitality, health, and mental component score compared to women. There were no differences when we evaluated the living environment of these patients, but we noted that people from the urban environment presented a higher score of general health, social function, and mental component compared to those from the rural environment (Table 5).

When comparing the two groups of patients, we observe that physical and social function, pain, general health, as well as the physical and mental components have much higher scores in the control group compared to those in the FH group.

With increasing age, we observe significant decreases in physical function, mental health, and the physical component, data that can be seen in Table 6. Although there is no statistical correlation, it is worth mentioning that the group aged over 65 has the lowest scores in all 10 components.

Using the Kruskal–Wallis test, we obtained the rejection of the null hypothesis between age and the physical function component (a *p*-value = 0.012 between those over 65 and under 45, and a *p*-value = 0.007 between those over 65 years and those in the 45–65 years group), general health (a *p*-value = 0.043 between those over 65 and those under 45, and a *p*-value = 0.003 between those over 65 and those in the 45–65 years), mental health (*p* = 0.043 between those over 65 and those under 45), as well as the physical component of the SF36 questionnaire (*p* = 0.032 between those over 65 and those in the group 45–65 years).

We note that in the control group, there were no differences in the evaluated age categories, but it is worth mentioning that the scores of patients over 65 years old were the lowest in the case of the physical component, social function, and emotional role (Table 7).

### 3.2. Eating Habits

We wanted to evaluate the emotional impact of eating, focusing on two aspects, which were component parts of the food frequency questionnaire: the causes of overeating and the effects determined by it.

We observed statistically significant differences in vitality and mental health in FH patients, with a higher score in patients who ate more due to loneliness. Among the patients who ate more out of boredom, no statistically significant differences were identified, but we note a lower somatic pain score and better physical and social function as well as better general and mental health compared to those who did not have this eating habit. Moreover, the physical and mental components were increased in patients who ate more out of loneliness and boredom than those who did not present this habit (Table 8).

No statistically significant differences were observed among patients who stated that they overeat when feeling upset. Still, we note that they present a lower physical and social function as well as a lower physical role compared to those who did not give this answer. Moreover, the scores given to the responses for general and mental health as well as for the somatic pain of these patients were higher among those who were overfed.

In the case of patients who motivated overeating through depression, the quality-of-life components had much lower scores. Values below 50 points were identified in the case of the physical role, somatic pain, general and mental health, as well as vitality. Statistically significant differences are highlighted in Table 9.

Regarding stress, no statistically significant differences were observed. Still, we observe that patients with FH who overeat due to a stressor present a better physical and social function as well as a better physical role, and a better mental component compared to those who did not state this.

Furthermore, the emotional role was statistically significantly influenced by the existence of social interactions and dining out. People who have lunch in society had a lower physical function score compared to those who did not declare this, but the physical role and social function were better. In addition, the physical and mental components of the quality-of-life score were higher in those who ate out, but without statistically significant differences (Table 10).

The patients included in the group of FH who eat more on weekends present a physical and social function as well as a physical and mental component with a better score compared to those who do not have this habit. Moreover, they present a more important somatic pain than those who did not declare this, but these results are without statistical significance (Table 11).

FH patients who overeat in front of the TV show lower scores in the following categories: physical role, somatic pain, social function, and physical component, but these differences are not statistically significant.

Assessing the control group, we noticed that there was not one person who ate more because of loneliness or depression. Moreover, in the control group, we observed higher quality-of-life component values in patients with FH (Table 12).

When we analyzed boredom as a determinant of overeating, we found significantly lower scores for somatic pain and general health compared to those who did not report this. We also observed lower scores for social function among the same patients, but the results were not statistically significant.

The number of those in the control group who overeat due to anger was low, but the scores obtained by them were clearly lower than the rest, a fact illustrated in Table 12.

Assessing people in the control group who overate due to stress, we observed that they had average values of quality-of-life components similar to those who had other reasons for overeating. Similar results were also observed in patients who overate in society, except for the mental component, where they had a slightly higher score than those who did not declare this (Table 13).

Comparing the group of patients with FH and the control group, we observe higher scores in almost all components of the quality-of-life questionnaire in the control group for overeating caused by stressful situations or eating out.

There were no statistically significant differences in patients who overate while sitting in front of the TV or during the weekend. However, we mention that vitality and mental health had better scores in patients from the control group who ate more in front of the TV compared to those who did not give this answer (Table 14).

Comparing the two groups, we observed data similar to those previously described, namely, those included in the control group had much higher values for all components of the quality-of-life score compared to those in the FH group.

Another aspect evaluated was the appearance of various reactions and feelings following overeating. We did not observe statistically significant differences in feelings of depression and guilt in the FH group (Table 15).

However, we note that patients who felt depressed after a large lunch had a better function and physical role but lower vitality, compared to those who did not experience these feelings. Similar results were observed for guilt, but they presented a better vitality score compared to those who did not give this answer.

Regarding the feeling of satisfaction produced by overeating, no statistically significant differences were observed, but those who experienced it had lower values of somatic pain, general health, and physical function. Patients with hypercholesterolemia who declared that they did not feel anything after a hearty lunch had statistically significantly lower values of somatic pain and vitality compared to those who did not give this answer (Table 16).

Moreover, those who gave up the next meal due to an exaggerated food intake had a significantly lower physical role compared to those who did not give this answer (Table 17). Also, patients who choose to exercise with the desire to balance overeating had the maximum physical role score.

No statistically significant differences were observed for patients who felt guilty after overeating. However, in their case, somatic pain had a greater impact on the quality of life, a fact also observed in the group of patients who felt a state of well-being (“satisfaction”) after a hearty meal. The same group of patients had a lower physical and mental component than those who did not give this answer, but the results are without significant correlations. In addition, the same group of patients had a clearly diminished social function, being statistically significant (Table 18).

Comparing the group of FH patients with the control group, we note that scores are higher for all components of the quality-of-life assessment questionnaire among the control subjects when assessing feelings of guilt and satisfaction after overeating.

We did not observe statistically significant differences in those who stated that overeating did not cause them any reaction, but we did note a low mean vitality score. Similar scores were calculated both for those who gave up the next meal and for the rest (Table 19).

Moreover, the subjects who declared that they exercised after an increased food intake had significantly higher scores of somatic pain, general health, and the physical component compared to those who did not exercise (Table 20).

Comparing the group of patients with FH with the control group, we noticed that people who declared that they exercise in the control group had much higher scores for all components of the quality-of-life score than those in the FH group. Similar results were observed for those who did not declare this.

## 4. Discussion

Familial hypercholesterolemia is a genetic condition with a strong impact both at the individual level (through the presence of arteriosclerosis, the increased risk of cardiovascular diseases, and death) and at the general (socio-economic) level. Numerous studies have considered therapeutic plans and the complications of FH but there are just a few dedicated to evaluating the quality of life in this category of patients.

Quality of life is a multidimensional concept defined from different individual perspectives, such as happiness, well-being, and satisfaction, from an intellectual and emotional standpoint. It is also considered to be good if patients can make their own decisions and have the ability to carry out various activities independently, maintaining their autonomy. Among the decisive factors that influence it is the state of health, but it can be defined as an individual perception of satisfaction in different areas considered important by each individual [9].

In the present study, both the overall scores and those obtained after the breakdown by quality-of-life components were lower in the FH group. Living with a high risk of developing a genetic disease can affect people’s lives in different ways. Awareness of personal risk can be one of the factors underlying a low quality of life. In addition, previous studies have shown that high cholesterol in adulthood is associated with a lower quality of life in the last few decades [10].

This explanation becomes even more plausible in the case of our study, since the average age of the patients in the study group is higher compared to the control group. Among all the components, somatic pain stands out, having the lowest score for those with FH. Older age and the presence of multiple comorbidities are two possible explanations.

Assessing the quality of life in patients with FH, we observed that women had better physical function and physical role scores. Still, the physical component was lower than that of the men included in the study group. Moreover, females showed lower social function and vitality compared to males but without statistically significant differences. For the control group, the results were similar to those of the FH patients, but we note that men had a significantly better social function and physical component than women.

Souto and his collaborators included, in a study, 658 patients who were suspected of having FH or were diagnosed with this disease and administered a quality-of-life questionnaire to them. The results described by the researchers demonstrated a decrease in the females’ physical and mental components, which are consistent with those observed in our study [11].

Similar results were also described by Mata and his collaborators, who performed a cohort study that included 1947 people, including 1321 patients with FH and 626 of their relatives. The authors evaluated the quality of life among the two groups, demonstrating a significant decrease in women and in the elderly [12].

Our study also quantified the area of origin of the patients, noting that those with FH from the rural environment presented a lower score for almost all components of the quality-of-life score compared to those from the urban environment, except the score obtained for physical function. A recently published study by Risal et al. aimed to assess the quality of life in a population of 439 subjects in Nepal. The authors demonstrated that the urban environment is a positive predictor of quality of life, and lower scores were obtained by patients from rural areas [13], these results being consistent with those of our study. The explanation as to why we can observe these effects in our population can be due to a deficient sanitary network in the rural environment or social deprivation (especially among the young or the elderly with multiple associated comorbidities), these being triggering factors for a decrease in the quality of life.

Furthermore, we observed statistically significant correlations between somatic pain in rural patients and a diminished quality of life. This was also described by Fujii et al., demonstrating in a cohort of 3100 patients that a more significant somatic pain caused a substantial decrease in quality of life, independent of the number of associated comorbidities [14]. However, the specialized literature published up to the present moment is poorly represented, and further studies including patients with FH are needed.

When it comes to age and quality of life, published studies show similar scores regardless of the age category, and some, on the contrary, note that young people have a lower quality of life. In a 2004 study, Torres et al. noted that the impact of cardiovascular disease on perceived health may not be as pronounced in older people as it is in younger people, a possible explanation being that health problems become more common with age, and these individuals may have adapted to their condition. Furthermore, older patients have lower expectations than younger individuals [15].

Hyttinen and his colleagues discuss, in their 2008 study, the “healthy survivor effect”, that is, those individuals who survive to old age even if they carry a genetic burden, such as FH, and who may have other positive protective factors, genetic or environmental. However, their research was conducted on a group of patients with a healthy, active lifestyle [16].

In the case of our study, the lowest scores occur in the FH patients over 65 years of age, with a score of 50 or even lower for half of the items. The control patients scored higher on all components, but even so, the geriatric patients scored lower. A possible explanation for this is that other factors negatively influence the quality of life, such as age, socio-economic status, or affective status.

Quantifying the eating habits of patients with FH, we observed that people who eat more in the context of loneliness or boredom present higher scores for most components of the quality-of-life score compared to those who do not have this habit. Santos and colleagues observed, in a recently published study, that uncontrolled overeating was associated with a significant decrease in vitality in patients with morbid obesity [17].

Hadar-Shoval and colleagues also assessed emotional eating and lifestyle changes during the COVID-19 pandemic. The authors divided the enrolled subjects into those who made positive or negative changes in their daily activities, as well as those who remained with the same habits before the appearance of the coronavirus. The researchers demonstrated that the patients who ate the most emotionally were those who had negative changes in the adopted lifestyle (fried foods and carbohydrates), followed by those who had positive changes (increased consumption of fruits and vegetables), and those who ate the least emotionally were the subjects who had the same habits as before the appearance of the coronavirus [18].

The differences observed in our study can be explained not only by differences in the associated chronic metabolic pathologies but also by cultural and geographical differences, underlining once again the multifactorial determinism of FH. Differences between our study and others may also be because our study enrolled patients both before and during the coronavirus pandemic.

In order to evaluate the correlations between stress and overeating, Costarelli and Patsai demonstrated, in a group of 60 female students from Greece, that under conditions of stress, 36.7% did not change their eating habits, 35% ate more, and 28.3% ate less [19]. Conversely, in a similar population of Chinese subjects, positive emotions were associated with an increased food intake compared to those experiencing negative emotions [20]. These results are consistent with our control group.

Thus, the results published in the specialized literature are contradictory and insufficient. Regarding the results of our study, we noticed that patients with FH who overeat for emotional reasons (loneliness, boredom, being upset, depression, or stress) present higher scores after completing the quality-of-life questionnaire compared to those who did not declare these behaviors. Possible explanations for these results can be considered the following:-Emotional overeating creates a generally good perception, amplifying satisfaction and implicitly the quality of life.-The general state of well-being is actually not due to emotional nutrition but to the correct prescription and its increased compliance with the appropriate treatment.-Some of the patients with FH are not aware of the fact that emotional overeating increases the cardiovascular risk through the appearance of obesity, diabetes, and metabolic syndrome, and they appreciate that their state of health is good.-In moments of psycho-emotional overload, it is possible for patients to resort to overeating, and then the patient returns to a healthy lifestyle, which overall will provide an increased quality of life score and, respectively, a better state of health.-The good quality of life in patients with FH can be an element that draws attention to the fact that many of the coexisting pathologies can be reversible by correcting modifiable risk factors (obesity and insulin resistance).

Moreover, our study assessed the feelings experienced after a large lunch. Jeong and Seo assessed satisfaction with food-related aspects of life in a group of patients. They demonstrated a positive correlation between food satisfaction and the overall quality of life [21].

In our study, we observed that FH patients who felt satisfaction after overeating presented lower scores of quality-of-life components compared to those who did not report this feeling. And in the control group, we observed the same tendency but with better scores compared to those with FH. A possible explanation can be given by the continuation of overeating due to the feeling of satisfaction, which will cause an increase in BMI, the appearance of metabolic syndrome, and all other complications associated with obesity, which will lead to a decrease in the quality of life.

FH patients who reported feeling depressed after overeating had a better physical component and a lower mental component than those who did not report this. Gonzales and his collaborators demonstrated, in a group of 180 patients, that subjects diagnosed with eating disorders presented a lower mental health, emotional role, and vitality compared to healthy individuals [22], results similar to those obtained in our study. We can thus claim that patients who experience depression as a result of food excesses have a better physical component compared to the mental one and a better vitality, these being low due to the depressive condition.

Regarding the patients with FH who gave up the next meal after an excessive food intake, we noticed that most scores in the quality-of-life score were lower in those with this habit than the rest of the patients. Ferrer-Cascales and his collaborators evaluated the correlations between eating or not eating breakfast and its quality with scores obtained when completing the quality-of-life questionnaire. The researchers demonstrated that the best scores were obtained by subjects who ate a high-quality breakfast, followed by those who did not eat the first meal of the day, and the lowest scores were obtained by individuals who ate a low-quality breakfast [23].

Furthermore, Tahara and his collaborators evaluated the correlations between eating meals irregularly and mental health in a group of Japanese subjects. The researchers showed that irregular lunches were associated with a decrease in the physical component, decreased work productivity, and an impaired sleep quality, possibly due to the lack of breakfast in the daily diet, an increased frequency of snacks and nutritional imbalances, as well as insufficient periods between the last meal and sleep [24]. In the context of overwork, stress, and excessive work, food and implicitly overeating bring with it, momentarily, a feeling of well-being. Awareness of the risk causes patients to skip the next meal, but over time, repeating this process can generate a feeling of frustration, the appearance of metabolic syndrome, and a decrease in quality of life.

Herrera-Espineira and colleagues assessed the differences in eating habits, physical activity, and quality of life in obese and overweight patients. These authors demonstrated that overweight patients had better indicators of a healthy diet, emotional eating, and physical activity when compared to obese patients [25]. In our study, patients with FH who chose physical activity after a food binge had better scores in almost all components of quality of life compared to those who were sedentary, results that are consistent with data from the literature.

To our knowledge, this is the first study in Romania that evaluates the quality of life of patients with FH, both in adults and in the geriatric population. The strength of this study is that we included the food frequency questionnaire besides the quality-of-live evaluation.

## 5. Conclusions

Overeating was driven by boredom and was more frequent on weekends in the FH group. None of the patients in the control group felt loneliness or depression associated with overeating, a possible explanation being the lower average age of this group. Half of the control group overeat during stressful times, on weekends, or in society. Emotional eating is a trigger for nutritional imbalances and obesity.

Regarding the patients with FH who skipped the next meal after an excessive food intake, we noticed that most scores were lower in those who had this habit compared to the rest of the patients, emphasizing the emotional component of this habit.

In the present study, both the overall scores and those obtained in each component were lower in the FH group. Lower scores were obtained by women, patients from rural living areas, and those over 65 years of age.

In conclusion, overeating in patients with FH is associated with a lower quality of life. The complexity of these patients needs a multidisciplinary approach; thus, the quality-of-life questionnaire should be implemented in their periodical follow-ups in order to increase their general status, with special attention paid to geriatric patients.

## Figures and Tables

**Figure 1 nutrients-15-03666-f001:**
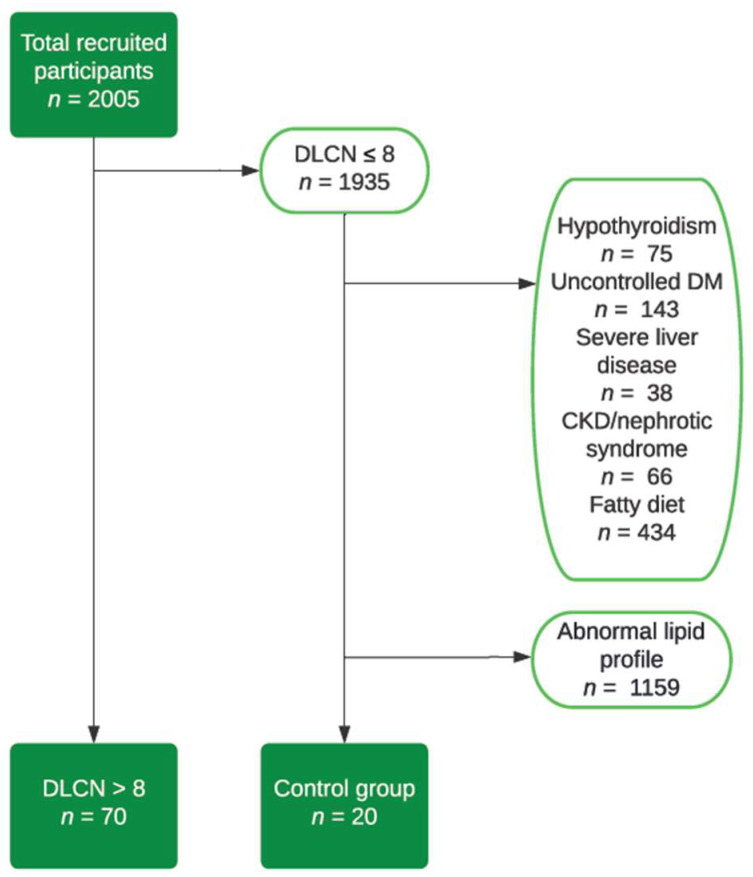
Flow chart—study group selection.

**Table 1 nutrients-15-03666-t001:** General characteristics of the study group.

General Characteristics	FH (n = 70)	Control (n = 20)	*p*
Gender n (%)			
Male	29 (41.4%)	7 (35.0%)	0.796
Female	41 (58.6%)	13 (65.0%)	
Living environment, n (%)			
Rural	18 (25.7%)	6 (30.0%)	0.776
Urban	52 (74.3%)	14 (70.0%)	
Age, years (mean ± SD)	54.65 ± 12.81	41.70 ± 13.51	<0.001

**Table 2 nutrients-15-03666-t002:** Quality-of-life assessment.

SF-36	FH	Control	*p*
Physical component	73.06 (54.67;83.42)	87.62 (82.68;91)	<0.001
Mental component	75.95 (60.20;85.60)	83.10 ± 13.07	0.004

**Table 3 nutrients-15-03666-t003:** Dimensions of the quality-of-life assessment score.

SF-36	FH	Control	*p*
Physical function	75 (53.75;95)	95 (77.5;100)	0.003
Role functioning/physical	75 (0;100)	100 (100;100)	0.001
Pain	52 (41;76.5)	84 (72.5;84)	0.002
General health	62 (45;77)	81.35 ± 12.14	<0.001
Vitality	65 (50;80)	72.5 ± 16.58	0.083
Social functioning	75 (62.5;100)	100 (87.5;100)	0.014
Role functioning/emotional	100 (66.6;100)	100 (100;100)	0.249
Emotional well-being	76 (60;88)	80.4 ± 10.61	0.147

Mean ± standard deviation or median (25th, 75th percentile), as appropriate.

**Table 4 nutrients-15-03666-t004:** Quality-of-life components and demographics of the FH cohort.

SF-36	Gender			Living Area		
	Female	Male	*p*	Rural	Urban	*p*
Physical function	85 (42.50;95)	75 (55;90)	0.947	85 (58.75;95)	72.5 (46.25;95)	0.512
Role functioning/physical	100 (25;100)	75 (0;100)	0.513	62.5 (0;100)	100 (25;100)	0.295
Pain	51 (36.50;79)	61 (51;79)	0.331	41 (22;54.5)	61.5 (51;84)	0.002
General health	62 (44.5;73.5)	67 (45;77)	0.299	56 (41.5;68.25)	67 (47;77)	0.161
Vitality	65 (47.50;75)	70 (52.50;70)	0.254	65 (53.75;75)	65 (50;80)	0.845
Social functioning	75 (62.50;100)	87.5 (62.50;100)	0.372	75 (62.5;87.5)	87.5 (62.5;100)	0.360
Role functioning/emotional	100 (66.60;100)	100 (83.30;100)	0.792	100 (33.3;100)	100 (74.95;100)	0.746
Emotional well-being	76 (56;86)	76 (64;88)	0.491	66 (59;81)	80 (60;88)	0.146
Physical component	69.25 (51.44;83.87)	75.31 (58.23;83.59)	0.617	64.79 (53.20;76.26)	75.59 (55.40;84.98)	0.093
Mental component	73.60 (56.49;84.56)	77.32 (65.20;86.30)	0.417	73.45 (55.92;79.85)	78.36 (61.15;87.95)	0.141

**Table 5 nutrients-15-03666-t005:** Quality-of-life components and control group demographics.

SF-36	Gender			Living Area		
	Female	Male	*p*	Rural	Urban	*p*
Physical function	95 (75;100)	95 (95;100)	0.384	95 (85;100)	95 (75;100)	0.863
Role functioning/physical	100 (100;100)	100 (100;100)	0.286	100 (100;100)	100 (100;100)	0.342
Pain	84 (67;84)	84 (84;100)	0.080	84 (63;88)	84 (69.5;88)	0.696
General health	81.53 ± 13.25	81 ± 10.73	0.928	85.16 ± 13.24	79.71 ± 11.76	0.372
Vitality	68.84 ± 16.72	79.28 ± 15.11	0.186	74.16 ±17.44	71.78 ± 16.82	0.777
Social functioning	87.5 (75;100)	100 (100;100)	0.012	93.75 (75;100)	100 (84.37;100)	0.640
Role functioning/emotional	100 (87.5;100)	100 (100;100)	0.180	100 (75;100)	100 (100;100)	0.791
Emotional well-being	77.84 ± 11.14	85.14 ± 8.23	0.147	81.33 ± 10.32	80.00 ± 11.09	0.805
Physical component	84.37 (78.5;90.18)	90.25 (86.37;91.87)	0.047	89.53 (76.89;92.64)	85.68 (82.04;91.40)	0.650
Mental component	79.87 ± 14.76	89.08 ± 6.35	0.137	81.88 ± 18.37	83.62 ± 10.90	0.793

**Table 6 nutrients-15-03666-t006:** Quality-of-life components by age groups in the FH cohort.

SF-36	Under 45	45–65	Over 65	*p*
Physical function	92.50 (56.25;98.75)	85 (55;95)	52.5 (22.5;68.75)	0.004
Role functioning/physical	87.5 (0;100)	100 (50;100)	50 (0;93.75)	0.114
Pain	67.5 (43.5;100)	52 (41;84)	51 (19.25;63.5)	0.194
General health	67 (44.25;85.75)	67 (53.75;77)	45 (29;52)	0.003
Vitality	67.5 (51.25;78.75)	67.5 (50;80)	62.5 (42.5;65)	0.242
Social functioning	87.5 (62.5;100)	87.5 (62.5;100)	75 (40.62;87.5)	0.262
Role functioning/emotional	100 (74.95;100)	100 (100;100)	100 (33.3;100)	0.403
Emotional well-being	86 (69;91)	76 (60;85)	66 (40;80)	0.050
Physical component	75.68 (53.43;91.03)	75.59 (63.05;84.35)	59.1 (38.37;73.85)	0.032
Mental component	79.75 (56.8;88.85)	77.36 (63.1;86.2)	61.56 (44.59;80)	0.060

**Table 7 nutrients-15-03666-t007:** Quality-of-life components by age groups in the control group.

SF-36	Under 45	45–65	Over 65	*p*
Physical function	95 (95;100)	95 (75;100)	77.5 (55;100)	0.893
Role functioning/physical	100 (100;100)	100 (100;100)	100 (100;100)	0.666
Pain	84 (73;92)	74 (73;84)	50 (0;100)	0.581
General health	77 (72;87)	87 (82;87)	88.5 (77;100)	0.553
Vitality	75 (65;80)	70 (70;80)	75 (50;100)	0.972
Social functioning	100 (87.5;100)	100 (87.5;100)	68.75 (37.5;100)	0.774
Role functioning/emotional	100 (100;100)	100 (100;100)	50 (0;100)	0.223
Emotional well-being	80 (80;84)	84 (80;84)	84 (68;100)	0.829
Physical component	89.75 (86.25;90.25)	84.37 (82.18;85.12)	74.21 (48.43;100)	0.637
Mental component	86.4 (81.4;88.5)	85.2 (82.7;87.4)	73.25 (46.5;100)	0.971

**Table 8 nutrients-15-03666-t008:** Components of quality of life and factors determining overnutrition in the FH group.

SF-36 FH Group	What Situations Make You Eat More?					
	Loneliness		*p*	Boredom		*p*
	YES	NO		YES	NO	
Physical function	87.5 (57.5;91.25)	70 (51.25;95)	0.606	85 (55;95)	70 (47.5;95)	0.480
Role functioning/physical	75 (50;100)	87.5 (0;100)	0.625	75 (25;100)	100 (0;100)	0.949
Pain	61.5 (51;84)	51.5 (41;74)	0.308	52 (41.5;84)	62 (41;74)	0.810
General health	67 (56.5;77)	59.5 (42;77)	0.225	67 (53.5;77)	57 (42;76)	0.131
Vitality	77.5 (61.25;85)	65 (46.25;75)	0.045	75 (52.5;82.5)	65 (40;75)	0.051
Social functioning	87.5 (71.87;100)	75 (62.5;100)	0.285	87.5 (62.5;100)	75 (50;100)	0.251
Role functioning/emotional	100 (91.65;100)	100 (66.6;100)	0.497	100 (100;100)	100 (49.95;100)	0.108
Emotional well-being	84 (71;88)	76 (57;84)	0.050	80 (62;88)	76 (52;84)	0.238
Physical component	75.88 (64.79;84.28)	69.43 (51.56;83.18)	0.191	75.81 (62.95;84.03)	67.87 (50.56;83.12)	0.289
Mental component	81.35 (69.72;88.9)	74.55 (54.1;85.17)	0.140	79.4 (67.05;87.4)	73.52 (51.58;84.4)	0.156

**Table 9 nutrients-15-03666-t009:** Components of quality of life and factors determining overnutrition in the FH group.

SF-36 FH Group	What Situations Make You Eat More?					
	Sorrow		*p*	Depression		*p*
	YES	NO		YES	NO	
Physical functioning	65 (40;90)	80 (55;95)	0.245	60 (13.75;90)	77.5 (55;95)	0.169
Role functioning/physical	75 (0;100)	100 (25;100)	0.394	50 (0;100)	87.5 (25;100)	0.362
Role functioning/emotional	100 (66.6;100)	100 (66.6;100)	0.954	100 (0;100)	100 (100;100)	0.125
Energy/fatigue	65 (45;85)	65 (50;75)	0.403	40 (13.75;65)	65 (55;80)	0.011
Emotional well-being	84 (60;88)	72 (60;84)	0.105	44 (19;78)	80 (64;88)	0.013
Social functioning	75 (50;100)	87.5 (62.5;100)	0.501	62.5 (12.5;87.5)	87.5 (62.5;100)	0.027
Pain	62 (51;84)	51 (41;74)	0.232	46 (0;67.5)	56.5 (41;81.5)	0.156
General health	72 (42;77)	57 (45;75)	0.301	41 (18.75;77)	64.5 (47.75;77)	0.088
Physical component	75.31 (55.12;83.93)	69.62 (53.31;83.25)	0.837	57.4 (10.09;79.34)	74.46 (57.32;83.76)	0.092
Mental component	77.32 (61.3;88.8)	74 (58.96;83.9)	0.500	60.15 (13.9;78.62)	77.36 (61.52;85.6)	0.038

**Table 10 nutrients-15-03666-t010:** Quality-of-life components and factors determining overnutrition in the FH group.

SF-36 FH Group	What Situations Make You Eat More?					
	Stress		*p*	In Society		*p*
	YES	NO		YES	NO	
Physical functioning	85 (51.25;95)	70 (53.75;95)	0.637	65 (45;100)	85 (55;95)	0.677
Role functioning/physical	100 (75;100)	62.5 (0;100)	0.014	100 (50;100)	75 (0;100)	0.549
Role functioning/emotional	100 (100;100)	100 (66.6;100)	0.232	100 (100;100)	100 (66.6;100)	0.047
Energy/fatigue	65 (51.25;75)	65 (50;80)	0.746	65 (60;80)	65 (50;80)	0.697
Emotional well-being	82 (60;88)	76 (59;85)	0.433	76 (64;88)	76 (56;88)	0.429
Social functioning	87.5 (53.12;100)	75 (62.5;100)	0.766	87.5 (62.5;100)	75 (62.5;100)	0.399
Pain	57 (43.5;100)	51 (41;74)	0.347	64 (51;100)	51 (41;74)	0.146
General health	62 (51.25;77)	62 (42;77)	0.476	67 (52;87)	62 (42;77)	0.162
Physical component	75.56 (64.7;86.09)	67.81 (50.84;82.96)	0.274	74.37 (62.25;86.81)	70.5 (51.12;83)	0.330
Mental component	77.35 (64.3;87.6)	73.8 (56.62;85)	0.656	80.2 (66.9;88.5)	74 (53.4;83.9)	0.167

**Table 11 nutrients-15-03666-t011:** Components of quality of life and factors determining overnutrition in the FH group.

SF-36 FH Group	What Situations Make You Eat More?					
	In Weekend		*p*	Watching TV		*p*
	YES	NO		YES	NO	
Physical functioning	80 (57.5;97.5)	70 (42.5;92.5)	0.240	70 (38.75;91.25)	77.5 (55;95)	0.575
Role functioning/physical	75 (12.5;100)	100 (0;100)	0.823	75 (18.75;100)	100 (0;100)	0.623
Pain	62 (46;84)	51 (36.5;69)	0.150	51 (29.5;84)	56.5 (41;74)	0.589
General health	67 (43.5;79.5)	62 (47;77)	0.793	64.5 (44.25;72.75)	62 (45.5;77)	0.973

**Table 12 nutrients-15-03666-t012:** Components of quality of life and factors determining overeating in the control group.

SF-36 Control Group	What Situations Make You Eat More?					
	Boredom		*p*	Anger		*p*
	DA	NU		DA	NU	
Physical functioning	92.5 (77.5;100)	95 (80;100)	0.805	77.5 (55;100)	95 (85;100)	0.693
Role functioning/physical	100 (81.25;100)	100 (100;100)	0.276	100 (100;100)	100 (100;100)	0.628
Role functioning/emotional	100 (81.25;100)	100 (100;100)	0.594	50 (0;100)	100 (100;100)	0.105
Energy/fatigue	65 ± 22.73	74.37 ± 15.04	0.325	72.5 (50;95)	72.5 (65;80)	1.000
Emotional well-being	74 ± 14.78	82 ± 9.23	0.185	80 (68;92)	80 (80;84)	0.897
Social functioning	87.5 (46.87;100)	100 (87.5;100)	0.391	68.75 (37.5;100)	100 (87.5;100)	0.475
Pain	61.5 (46;78.5)	84 (76.5;96)	0.037	30.5 (0;61)	84 (74;84)	0.024
General health	67 ± 12.24	84.93 ± 9.39	0.005	79.5 (77;82)	82 (72;92)	0.751
Physical component	80.87 (63.54;90.03)	89.31 (84.23;91.46)	0.298	69.84 (48.43;91.25)	87.62 (84.18;90.25)	0.614
Mental component	76.3 (59.72;90.7)	86.8 (83.17;90.52)	0.219	70.15 (46.5;93.8)	85.9 (81.4;88.5)	0.801

**Table 13 nutrients-15-03666-t013:** Components of quality of life and factors determining overeating in the control group.

SF-36 Control Group	What Situations Make You Eat More?					
	Stress		*p*	In Society		*p*
	YES	NO		YES	NO	
Physical functioning	95 (75;100)	95 (85;100)	0.751	95 (95;100)	95 (75;100)	0.451
Role functioning/physical	100 (100;100)	100 (100;100)	0.884	100 (100;100)	100 (100;100)	0.884
Role functioning/emotional	100 (100;100)	100 (100;100)	0.625	100 (100;100)	100 (100;100)	0.625
Energy/fatigue	72.72 ± 16.78	72.22 ± 17.34	0.948	73.33 ± 15.61	71.81 ± 18.06	0.845
Emotional well-being	81.45 ± 10.16	79.11 ± 11.62	0.636	80.88 ± 9.33	80 ± 12	0.858
Social functioning	100 (87.5;100)	100 (81.25;100)	0.730	87.5 (87.5;100)	100 (75;100)	0.546
Pain	84 (62;84)	84 (78;92)	0.447	84 (79;100)	84 (61;84)	0.101
General health	82.09 ± 13.01	80.44 ± 11.7	0.772	85.22 ± 9.64	78.18 ± 13.46	0.205
Physical component	85.12 (82.18;91.25)	88.87 (83.93;91.06)	0.621	89.75 (83.18;90.21)	86.37 (75.37;91.87)	0.820
Mental component	85.2 (77.9;92)	86.4 (78.8;89.85)	0.970	85.66 ± 6.92	81 ± 16.59	0.411

**Table 14 nutrients-15-03666-t014:** Components of quality of life and factors determining overeating in control patients.

SF-36 Control Group	What Situations Make You Eat More?					
	In Weekend		*p*	Watching TV		*p*
	YES	NO		YES	NO	
Physical functioning	95 (87.5;95)	97.5 (71.25;100)	0.600	92.5 (66.25;100)	95 (80;100)	1.000
Role functioning/physical	100 (81.25;100)	100 (100;100)	0.075	100 (100;100)	100 (100;100)	0.468
Role functioning/emotional	100 (81.25;100)	100 (100;100)	0.385	100 (100;100)	100 (100;100)	0.361
Energy/fatigue	77.5 (65;80)	70 (60;91.25)	1.000	82.5 ± 17.55	70 ± 15.91	0.184
Emotional well-being	77 ± 9.97	82.66 ± 10.83	0.253	84 ± 15.66	79.5 ± 9.45	0.463
Pain	84 (65;84)	84 (72.5;96)	0.685	73 (61.25;96)	84 (74;84)	0.654
General health	79.87 ± 14.3	82.33 ± 11.03	0.670	80.25 ± 20.3	81.62 ± 10.19	0.846
Physical component	89.31 (77.07;91.45)	86.31 (84.23;91)	0.847	88.18 (77.81;97.81)	87.62 (82.68;90.23)	0.670
Mental component	85.5 (74.07;90.52)	86.3 (78.77;92.47)	0.671	90.6 (75.25;98.45)	85.3 (78.77;88.5)	0.277

**Table 15 nutrients-15-03666-t015:** Quality-of-life components and response to overeating in the FH group.

SF-36 FH Group	How Do You Feel after Overeating?					
	Depression		*p*	Guilt		*p*
	YES	NO		YES	NO	
Physical functioning	85 (25;90)	75 (55;95)	0.701	85 (55;96.25)	72.25 (46.25;95)	0.380
Role functioning/physical	100 (50;100)	75 (0;100)	0.260	100 (43.75;100)	75 (0;100)	0.115
Pain	62 (52;100)	51 (41;74)	0.222	62 (46.25;84)	51 (41;74)	0.415
General health	62 (42;72)	62 (45;77)	0.624	64.5 (47;79.5)	59.5 (42;77)	0.265

**Table 16 nutrients-15-03666-t016:** Quality-of-life components and response to overeating in the FH group.

SF-36 FH Group	How Do You Feel after Overeating?					
	Satisfaction		*p*	Nothing		*p*
	YES	NO		YES	NO	
Physical functioning	65 (37.5;90)	80 (55;90)	0.373	90 (55;95)	70 (50;90)	0.430
Role functioning/physical	75 (0;100)	75 (12.5;100)	0.713	75 (0;100)	100 (25;100)	0.398
Role functioning/emotional	100 (100;100)	100 (66.6;100)	0.277	100 (66.6;100)	100 (66.6;100)	0.993
Energy/fatigue	65 (50;72.5)	65 (50;80)	0.646	65 (40;65)	70 (55;80)	0.048
Pain	51 (26.5;73)	61 (41;79)	0.345	51 (32;61)	62 (51;84)	0.020
General health	57 (38.5;77)	62 (46;77)	0.450	67 (50;77)	62 (42;77)	0.696
Social functioning	62.5 (50;93.75)	87.5 (62.5;100)	0.208	75 (62.5;100)	75 (62.5;100)	0.908
Emotional well-being	72 (60;88)	76 (60;88)	0.909	80 (56;88)	76 (60;88)	0.926
Physical component	64.4 (47.45;79.71)	74.56 (54.76;83.59)	0.400	69.62 (53.31;77.95)	74.56 (55.12;84.12)	0.484
Mental component	67.1 (59.25;80.55)	77.32 (59.79;86.3)	0.505	73.6 (61.1;82.5)	77.3 (58.96;87.8)	0.634

**Table 17 nutrients-15-03666-t017:** Quality-of-life components and response to overeating in the FH group.

SF-36 FH Group	How Do You Feel after Overeating?					
	I Skip the Next Meal			I Exercise		
	YES	NO	*p*	YES	NO	*p*
Physical function	65 (50;85)	85 (55;95)	0.222	75 (60;90)	75 (45;95)	0.685
Role functioning/physical	50 (0;100)	100 (25;100)	0.043	100 (50;100)	75 (0;100)	0.179
Pain	62.5 (48.5;74)	51.5 (41;84)	0.593	61 (51;84)	52 (32;74)	0.182
General health	57 (44.25;73.25)	64.5 (47;77)	0.604	55 (43.5;87)	62 (46;73.5)	0.658
Energy/fatigue	65 (58.75;80)	65 (50;80)	0.660	75 (50;80)	65 (50;75)	0.154
Social functioning	75 (50;90.62)	87.5 (62.5;100)	0.339	87.5 (62.5;100)	75 (62.5;100)	0.486
Role functioning/emotional	100 (91.65;100)	100 (66.6;100)	0.726	100 (100;100)	100 (66.6;100)	0.418
Emotional well-being	76 (62;88)	76 (60;88)	0.952	76 (66;88)	76 (58;88)	0.571
Physical component	69.18 (47.58;78.60)	74.93 (57.29;84.59)	0.295	79.2 (62.95;85.78)	69.62 (52;81.06)	0.209
Mental component	73.7 (56.62;84.67)	76.8 (60.74;85.6)	0.702	80.2 (60.03;88.95)	74 (58.69;83.05)	0.254

**Table 18 nutrients-15-03666-t018:** Quality-of-life components and response to overeating in the control group.

SF-36 Control Group	How Do You Feel after Overeating?					
	Guilt			Satisfaction		
	YES	NO	*p*	YES	NO	*p*
Physical function	95 (87.5;98.75)	95 (75;100)	0.921	85 (70;90)	95 (95;100)	0.109
Role functioning/physical	100 (81.25;100)	100 (100;100)	0.276	100 (100;100)	100 (100;100)	0.542
Pain	68 (61.25;93.5)	84 (76.5;84)	0.320	62 (31;73)	84 (74;84)	0.184
General health	82 (59.5;85.75)	79.5 (73.25;94.25)	0.634	74.66 ± 21.59	82.52 ± 15.16	0.314
Energy/fatigue	72.5 (65;91.25)	72.5 (60;80)	0.775	65 ± 15	73.82 ± 16.91	0.410
Social functioning	93.75 (78.12;100)	100 (87.5;100)	0.748	75 (56.2;81.25)	100 (87.5;100)	0.016
Role functioning/emotional	100 (81.25;100)	100 (100;100)	0.594	100 (50;100)	100 (100;100)	0.268
Emotional well-being	80 (67;90)	80 (80;84)	0.846	70.66 ± 8.32	82.11 ± 10.2	0.085
Physical component	85.96 (77.07;90.87)	87.62 (84.23;91.46)	0.570	75.37 (61.90;82.78)	88.87 (84.37;91.25)	0.138
Mental component	83.65 (74.07;91.5)	86.8 (78.77;90.52)	0.570	71.2 (58.85;79.85)	86.4 (82.7;91.2)	0.168

**Table 19 nutrients-15-03666-t019:** Quality-of-life components and response to overeating in the control group.

SF-36 Control Group	How Do You Feel after Overeating?					
	Nothing			I Skip the Next Meal		
	YES	NO	*p*	YES	NO	*p*
Physical function	95 (75;100)	95 (80;100)	0.868	95 (80;100)	95 (77.5;100)	0.778
Role functioning/physical	100 (100;100)	100 (100;100)	0.648	100 (100;100)	100 (100;100)	0.767
Pain	84 (72;84)	84 (68;92)	1.000	84 (76.5;96)	84 (61.25;84)	0.255
General health	75.57 ± 7.48	84.46 ± 13.25	0.121	86.5 ± 9.3	77.91 ± 12.94	0.124
Energy/fatigue	67.14 ± 15.23	75.38 ± 17.13	0.301	73.75 ± 15.75	71.66 ± 17.75	0.791

**Table 20 nutrients-15-03666-t020:** Quality-of-life components and response to overeating in the control group.

SF-36 Control Group	How Do You Feel after Overeating?		*p*
	I Exercise		
	YES	NO	
Physical function	100 (97.5;100)	95 (75;100)	0.167
Role functioning/physical	100 (100;100)	100 (100;100)	0.542
Pain	100 (100;100)	84 (67;84)	0.007
General health	100 (91;100)	77 (72;87)	0.048
Energy/fatigue	100 (82.5;100)	70 (60;80)	0.134
Social functioning	100 (100;100)	100 (87.5;100)	0.149
Role functioning/emotional	100 (100;100)	100 (100;100)	0.443
Emotional well-being	100 (88;100)	80 (80;84)	0.174
Physical component	100 (94.87;100)	86.25 (82.18;90.18)	0.039
Mental component	100 (92.3;100)	85.40 (77.90;88.5)	0.101

## Data Availability

Not applicable.

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
