# Peer review of "Association between Eating Patterns and Quality of Life in Patients with Familial Hypercholesterolemia"

_nutrients, 2023, doi:10.3390/nu15163666_

Round 1

Reviewer 1 Report

dear authors,

The paper about eating patterns , quality of life and familial hypercholesterolemia has provided a sufficient background and relevant references? The cited references are relevant to the research and the research design is appropriate

The methods adequately described,

except the description of the patient procedures selection;

How did you exclude the ones with fatty diet – what is the definition of a fatty diet

The total sum does not equal 1159 excluded ones but less then 700 ?

In methodology explain from physical function, vitality,…,  to mental component- from which questions did they came out – all variables ?

You can give the questionnaire in appendix or at the last section of the paper.

The references should be created in MDPI style, bold, italics fonts, use the authors’ manual / paper template.  

Author Response

Dear reviewer,

Thank you for all your comments.

  1. We have added more references to our paper. Thank you!
  2. We have excluded the patients with fatty diet according to the ESC guidelines for the management of dyslipidemias from 2019. In the guidelines it is stipulated that a fatty diet is considered as the total fat intake more than 40% of the daily caloric intake.
  3. Regarding the flow chart, we have firstly evaluated the DLCN score. If the patients had a score more than 8, they were considered as having FH. For the patients with a score less than 8, we found only 20 subjects without chronic diseases and with normal lipid profile values. Hopefully the new flow chart is more clear.
  4. Regarding the variables included in our study, these are the terms used in the unique interpretation recommended by the authors of the SF-36 Questionnaire. We have added in the Supplementary Materials the questionnaire in the English version. The interpretation of the questionnaire can be found at the following link: https://www.rand.org/health-care/surveys_tools/mos/36-item-short-form/scoring.html
  5. We have modified the references in MDPI style.

Hope we have touched all the points you asked us to change.

If there are any other changes you consider we should make, please let us know.

Yours sincerely,

All the authors

Reviewer 2 Report

Thank you for submitting the manuscript "Association between eating patterns and quality of life in patients with familial hypercholesterolemia" to Nutrients.

Line#21: Reading this sentence it seems that only for these families is food important. It's not because it's a short sentence that it needs to be poorly explained.

Line#25: Give more information about the control group.

Line#26: eating habits of families in general or family members separately?

Add numerical values and the probability of the statement you are making to be true even in the abstract.

Line#28: scores of what?

Line#29: Your study completion is greater than the results item. Overall, the abstract needs to be improved.

Introduction: the item needs to be improved. It needs to contain the justification for carrying out this work, as a simple exposition of the results of the dietary patterns of the group is not a research work to be published. Why is it important to know this group of people? How do the eating patterns of people affected by hypercholesterolemia impact family eating patterns?

Material and Methods:

Line#53: In the period of almost two years only 70 patients used the clinic? Is this number the number of data that were collected or the number of the population that after passing the inclusion criteria resulted in 70 patients included in the survey? This needs to be made clear. Maybe bring the information from lines #62-64 into this part.

Line#76: the questionnaire must appear as a supplementary file.

Line#86: unite?

Results: the authors need to find a way to reduce the number of tables since this way it is difficult to read the results due to the large number they appear in the text.

Minor editing of English language required

Author Response

Dear reviewer,

Thank you for all your comments.

  1. We have added more details in the abstract, including the definition of familial hypercholesterolemia as well as numbers in the results section. Hopefully, this underlines the importance of the subject.
  2. As you mentioned, we have added more details regarding the control group. Thank you very much!
  3. In Line#28, we have completed. Thank you!
  4. We have added in the introduction section more details regarding familial hypercholesterolemia, including the definition, prevalence and pathogenesis.
  5. We hope that the improved flow chart will explain better the selection of the patients (from the total of 2005 patients, only 70 (3,49%) had familial hypercholesterolemia).
  6. We have added the SF-36 Questionnaire as a supplementary material. Thank you!
  7. We have resolved Line#86.
  8. Regarding the tables, all authors consider them important in the flow of the results, due to the comparison between the groups. Usually in the MDPI format, the table is minimized and doesn’t interfere with the reading of the article.

Hope we have touched all the points you asked us to change.

If there are any other changes you consider we should make, please let us know.

Yours sincerely,

All the authors

Round 2

Reviewer 1 Report

Dear authors 

the text was corrected according the review. It is now ready for publication